# Symptom improvement in adenomyosis patients after ultrasound guided microwave ablation or uterine artery embolization, A randomized controlled pilot study

Gudny Jonsdottir[1,2]*, Erika Lantz[1,2], Marie Beermann[2,3], Maria Paschou[1], Helena Kopp Kallner[1,2], Klara Hasselrot[1,2]

1 Department of Obstetrics and Gynecology Danderyd Hospital, Stockholm, Sweden, 2 Department of Clincial Sciences at Danderyd Hospital, Karolinska Institutet, Stockholm, Sweden, 3 Department of Radiology Danderyd Hospital, Stockholm, Sweden

* gudny.jonsdottir@ki.se

## Abstract

### Introduction

The objective of this study was to investigate alleviation of adenomyosis symptoms after microwave ablation (MWA) and uterine artery embolization (UAE) in a small pilot study.

### Material and methods

20 premenopausal women with symptomatic adenomyosis were included at Danderyd Hospital, Sweden, from June 2020 to February 2023. Patients were randomized to MWA or UAE. The primary outcome was symptom severity score (SSS) at 6 months post treatment evaluated through the UFS-QoL questionnaire. The secondary outcomes were comparison of: health related quality of life (HR-QoL/UFS-QoL), Pictorial Bleeding Assessment Chart (PBAC), dysmenorrhea on a numerical rating scale (NRS), uterine volume, hemoglobin, Ca-125, prolactin and Anti Müllerian hormone (AMH), duration of hospitalization, use of pain medication and acceptability. As exploratory outcomes, we evaluated postoperative pain and return to daily activities. Clinical trials number NCT04209127.

### Results

There was no significant difference in primary outcome between the groups: SSS decreased within the MWA group from 69 to 44 (p=0.007), and within the UAE group from 88 to 47 (p=0.067). Quality of life increased significantly in the MWA group from 27 to 79 (p=0.002) and in the UAE group from 13 to 67 (p=0.013). Dysmenorrhea decreased in both groups; NRS from 6 to 1 (p=0.008) in the MWA group and from 9 to 4 (p=0.02) in the UAE group. The MWA group had significantly shorter

**Data availability statement:** Data was not registered in a public database but is available at our research clinic in a local database and on paper. According to the General data protection regulation (GDPR) of the European union no data containing personal information can be shared outside the European Union unless patients have consented. Data can be shared within the European Union after establishment of an agreement. Data can therefore not be uploaded in a repository, nor disclosed within the manuscript or uploaded as supplementary information. According to the permission granted this study from our Regional Ethics board, all data is stored and available for 15 years. Data requests may thus be sent to the Data Protection Officer (DPO) of Danderyd Hospital, e-mail address: dso.ds@regionstockholm.se.

**Funding:** The study was funded by The Stockholm County Council. HKK received the funding, the sponsors were not involved in anything connected to the study." "The funders had no role in study design, data collection and analysis, decision to publish, or preparation of the manuscript".

**Competing interests:** All authors have declared that no competing interests exist.

**Abbreviations:** MWA, microwave ablation; UAE, uterine artery embolization; SSS, symptom severity score; UFS-QoL, uterine fibroid symptom and quality of life; PBAC, pictorial bleeding assessment chart; NRS, numeric rating scale.

hospitalization (0 days vs 3, p=0.004), and quicker return to daily activities (3 days vs 14, p=0.005), compared to the UAE group. No serious adverse events occurred.

## Conclusion

In this small pilot trial, we had low power to detect differences between groups. Both treatments resulted in a significant decrease of symptoms related to adenomyosis. Postoperative recovery seems superior after MWA, in line with previous trials. Further investigations regarding MWA for adenomyosis are needed.

## Introduction

Adenomyosis is a benign condition causing symptoms such as heavy menstrual bleeding (HMB), anemia, dysmenorrhea, chronic pelvic pain, and impaired fertility [1,2]. About 2/3 of patients with adenomyosis have symptoms that negatively impact their quality of life and lead to absence from work [3,4]. The pathogenesis of adenomyosis results in endometrial deposits within the myometrium leading to either focal or diffuse adenomyotic lesions [5,6]. Adenomyosis is considered diffuse when endometrial glands and stroma are spread in the myometrium and focal when they are isolated in one part of the myometrium [7,8]. Earlier reports of the prevalence of adenomyosis were based on hysterectomy specimens, which may not be accurate to describe the true burden of the disease [1,2,9]. With better awareness about adenomyosis and more consensus in diagnostic criteria on ultrasound and magnetic resonance imaging (MRI), the diagnosis can now be made by noninvasive methods [10–13]. Among women seeking care for symptoms such as menorrhagia and dysmenorrhea, the reported prevalence on transvaginal ultrasound has been between 21% to 34% [10]. When medical treatment fails, hysterectomy has often been the recommended treatment, which is curative for bleedings but might not reduce the need of pain medication [14]. For patients that want to preserve their uterus and to avoid risks of major surgery, accessibility of alternative treatment options varies.

As for uterine fibroids, minimally invasive treatment options for adenomyosis are receiving increased attention and ablative techniques are developing rapidly. Uterine artery embolization (UAE) is a well-established non-invasive treatment option for fibroids and adenomyosis [15]. A meta-analysis has shown that UAE improved clinical symptoms of patients with isolated adenomyosis in 89% of cases in the short term follow up (during the first 12 months after treatment) and 74% of cases in the long term follow up (12-65 months after treatment) [15]. Ablative techniques like Radio Frequency ablation (RF), High-intensity focused ultrasound (HIFU) and Microwave Ablation (MWA) have been studied as treatment of adenomyosis, with similar effect on symptoms and few adverse events, and with shorter ablation time for MWA [16–18]. Also, MWA treatment of adenomyosis has not shown any effect on ovarian reserve [19]. A review article from 2021 including 13 published studies and 736 patients showed symptom reduction and few mild complications after MWA of adenomyosis [20]. Uterine volume decreased by 55-65%, dysmenorrhea by 50-82%,

SSS by 21-60% and anemia by 56-79% at follow up after 3 to 12 months. To our knowledge, no previous RCT between MWA and UAE for adenomyosis has been performed, and patient's acceptability of MWA treatment for adenomyosis has not been reported. The effect of MWA treatment of adenomyosis have previously been monitored by Ca-125 [21] and prolactin [22] levels in blood, as greater uterine size has been correlated with higher Ca-125 levels [23]. Our earlier studies on MWA for uterine fibroids [24–26] have recently led to implementation of this treatment into clinical praxis at Danderyd Hospital.

The aim of this study was to evaluate the potential of efficacy and acceptability of MWA and UAE on symptomatic adenomyosis in a randomized controlled pilot trial.

## Materials and methods

The study was approved by the Swedish Ethical Review Authority with application number 2019-04262 and registered on clinicaltrials.gov with identifier NCT04209127. Patients were recruited at Danderyd Hospital in Stockholm, a tertiary teaching hospital affiliated to Karolinska Institutet. From June 8, 2020, to February 28, 2023, a total of 20 patients were included in the study. The small study size was chosen due to absent previous clinical experience at our hospital of MWA or UAE as treatments for isolated adenomyosis. No power calculation was made. Follow-up visits were scheduled two- and six-months post treatment.

The inclusion criteria's were: premenopausal status 30-55 years of age, symptoms of adenomyosis and having at least two ultrasound criteria of adenomyosis. The exclusion criteria's were: current or future pregnancy wish, Body Mass Index (BMI) ≥ 35, treatment with anticoagulant or having a bleeding disorder, verified or suspected endometriosis on ultrasound or MRI, contraindication to UAE or general anesthesia, fibroids >3 cm, and hormonal treatment three months prior to treatment. All patients were examined by a specialist in gynecological ultrasound. The ultrasound criteria used to identify adenomyosis were the following: a globular uterus or asymmetric uterine walls, fan shaped shadowing, heterogenous myometrium, ill-defined junctional zone, increased vertical/translesional vascularization and suspected adenomyosis cysts, as defined by the MUSA criteria described by Harmsen et al. [12]. Patients with focal and/or diffuse adenomyosis were included. If patients met the inclusion criteria and lacked exclusion criteria, they underwent an MRI as secondary imaging modality after signing informed consent. If adenomyosis was confirmed on MRI, the patients were included in the study and randomized 1:1 to either MWA or UAE by consecutive opening of numbered opaque sealed envelopes containing the allocation. The envelopes were prepared by staff not involved in the study following a computer-generated randomization list (www.sealedenvelope.com). The result of the randomization was not deemed possible to blind to the patients nor the providers.

The length, anterio-posterior (AP) and width of the uterus were measured on MRI prior to the study and at the six-month follow-up (Figs 1 and 2), to calculate uterine volume according to a defined formula (in centimeters: length x AP x width x 0.5233) [27]. Before treatment and at six-months follow-up all patients completed a validated questionnaire about symptoms and quality of life (Uterine fibroid symptom and health-related quality of life questionnaire, UFS-QoL) [28] and the validated Pictorial Blood loss Assessment Chart (PBAC) [29] to assess the menstrual blood loss. UFS-QoL generates two scores, a symptom severity score (SSS) where high scores indicate more severe symptoms, and a score on health-related quality of life (HR-QoL) where high scores indicate a higher quality of life. PBAC generates a score based on numbers of pads and tampons used, the degree of soaking of pads and tampons, clots passed and days of bleeding during a menstrual period. We also included numeric rating scale (NRS) [30] of maximum daily pain during menstruation (on a scale from 0-10) as a proxy for dysmenorrhea before treatment and at six-months follow-up. Patients also evaluated their maximum pain on NRS postoperatively and daily for the first 7 days at home after treatment. The hemoglobin count, Ca-125, prolactin and AMH were evaluated before treatment and at six months follow-up. A follow up MRI was performed six months after treatment (Figs 1 and 2). At the two-month follow-up length of hospitalization was registered and patients were asked about time to return to daily activities and postoperative pain medication. On treatment day and at six

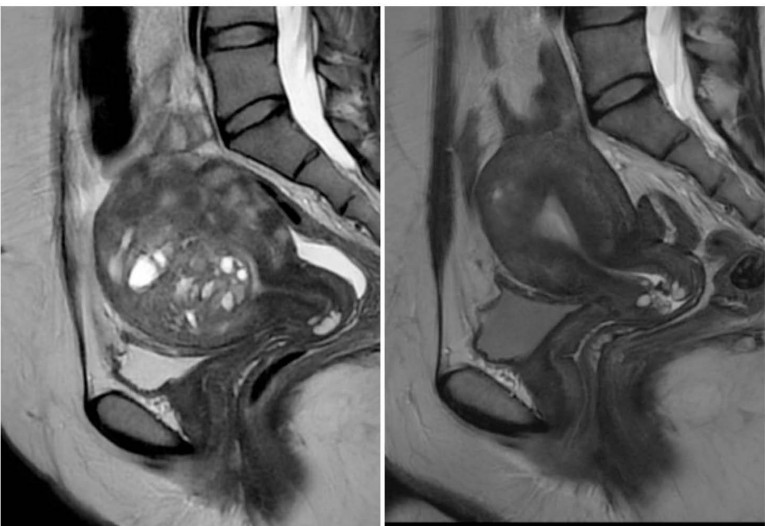

**Fig 1. Sagittal MRI without iv contrast of uterus with adenomyosis at baseline (left) and at 6 months post microwave ablation (right).**

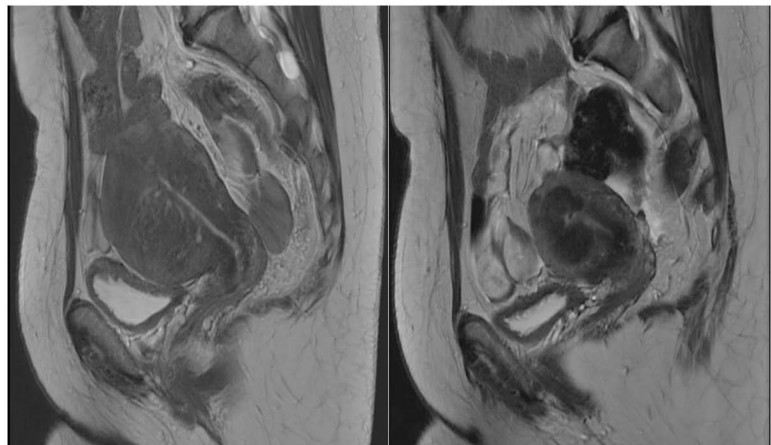

**Fig 2. Sagittal MRI without iv contrast of uterus with adenomyosis at baseline (left) and at 6 months post uterine artery embolization (right).**

months-follow up patients were asked if they would recommend the treatment to a friend using a Likert scale from 1 (very unlikely) to 7 (very likely) to assess acceptability. A score from 1-3 was considered as "would not recommend", score 4-5 as "would neither recommend nor discourage" and score 6-7 as "would strongly recommend".

An Emprint™ Ablation Generator with Thermosphere™ Technology (Medtronic, Minneapolis, USA) with a 13-gauge antenna was used for MWA. The generator operates at a frequency of 2.45 GHz ± 50 MHz with an output power range of 5–100 Watts. The MWA was performed transabdominally or vaginally in the operation room under general anesthesia in supine position with the presence of a gynecologist and an interventional radiologist. Ultrasound (LOGIQ™ 10, GE Healthcare) was used for imaging and needle guidance. Contrast enhanced ultrasound was used to assess size and vascularization of the adenomyosis, with the administration of 2.4 ml of SonoVue® medium (Bracco, Milan, Italy) before and immediately after MWA. The microwave effect in watts used was 50–100 Watts and the total time of ablation per patient

was between 5 and 31 minutes. The effect of the MWA on the adenomyosis was monitored by real-time ultrasound and the ablation was stopped when the microbubbles, that roughly represent the ablation zone, covered the adenomyosis affected area. A standardized protocol was used for perioperative pain management for all MWA patients, they were given pre-operative medication of 1.5 g paracetamol, 400 mg celecoxib, and 5 mg oxycodone. Post-operative pain was managed with intravenous morphine if needed.

The UAE was performed by an interventional radiologist at Danderyd Hospital under epidural anesthesia using sufentanil and ropivacaine according to hospital routine. UAE was carried out under fluoroscopic guidance (Alphenix™, INFX-8000V, Canon Medical System Corporation, Askim, Sweden). A unilateral groin approach was used for bilateral, selective embolization of the uterine artery. Through a microcatheter (Embocath® Plus, Merit Medical, Stockholm Sweden) tris-acryl polymer microspheres 500–700 mm (Embosphere®, Merit Medical, Stockholm, Sweden) were injected until flow in the artery ceased in one side and then the other side. All patients had urinary catheter to avoid urinary retention related to the epidural. Postoperative pain management was by a standardized protocol as follows: the day after UAE patients received 400 mg celecoxib, 1g paracetamol and 10 mg oxycodone, after which the epidural was turned off. If the pain was not tolerated by the patient, the epidural was turned on again using only ropivacaine. The procedure was repeated daily until the patient could leave the hospital with per oral analgesia.

Primary outcome was difference in symptoms six months after treatment, measured by the symptom severity score (SSS).

Secondary outcomes were changes in quality of life, amount of menstrual bleeding/PBAC, dysmenorrhea on NRS, uterine volume, levels of Hb, Ca-125, prolactin and AMH, hospitalization, use of postoperative pain medication and acceptability. As exploratory outcomes, we evaluated postoperative pain and return to daily activities after treatment.

Data generated or analyzed during the study are available from the corresponding author on request.

## Statistical analysis

Statistical Package for Social Sciences (SPSS) version 29 (IBM corporation, Armonk, New York, US) was used for statistical analysis. For baseline characteristics, comparison between the groups for continuous variables, and for analysis of differences between the groups for variables that were measured before and after treatment we used the Mann Whitney U-test. Statistical significance was assumed at a p-value <0.05. Median of the difference in outcomes between the groups and 95% Confidence intervals were calculated by custom tables. To compare the difference between variables that were measured before and after treatment within each group we used Paired T test and descriptive statistics. Acceptability of treatment was categorized into three groups "would recommend" (6–7), "would neither recommend nor discourage " (4–5) and "would not recommend" (12–3). Differences between groups were analyzed by Chi2 test. We tried to apply multiple linear regression analysis on baseline variables, but the model lacked sufficient explanatory power.

## Results

Study flow is shown in the consort flow chart (Fig 3). Of the 20 randomized patients, 10 were randomized to MWA and 10 to UAE. During the study period eight patients were treated with MWA and seven with UAE. The MWA treatment was performed by abdominal approach in 6/8 cases and vaginally in 2/8 cases. Baseline characteristics are presented in Table 1. There was no difference between the groups regarding preoperative SSS, PBAC score, HR-QoL score, age, BMI, parity, hemoglobin, prolactin, AMH, or dysmenorrhea. Despite randomization, the MWA group had significantly higher Ca-125 at baseline.

Primary and secondary outcomes are presented in Table 2. The MWA group had a significant decrease in SSS of 36%, median 69 to 44 (p=0.007), while the UAE group decreased in median SSS by 47% from 88 to 47 (p=0.067). The decrease

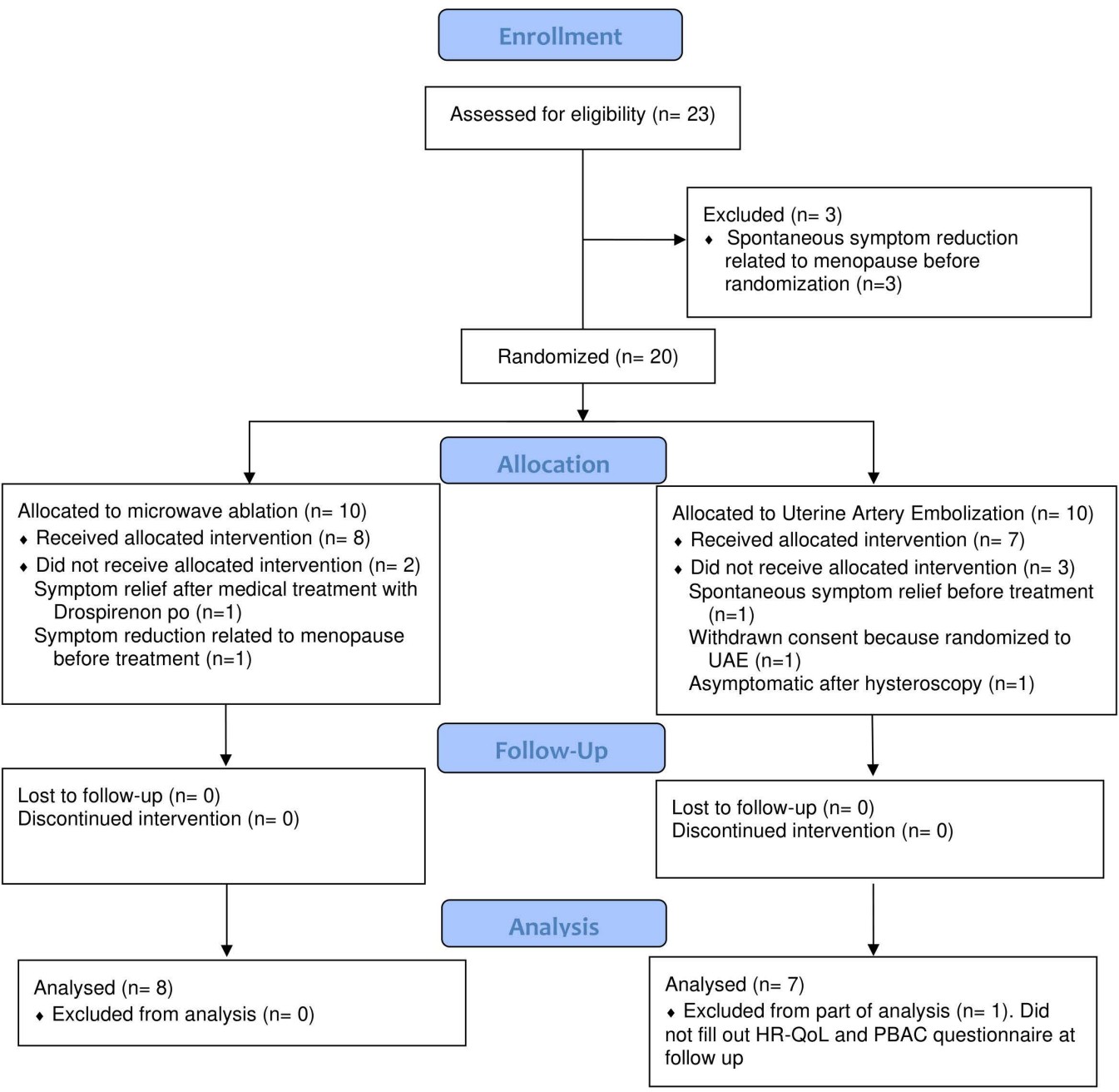

**Fig 3. Consort Flow Chart.**

in SSS between the two groups was not significant. Quality of life score increased significantly in the MWA group (p=0.002) and in the UAE group (p=0.013) at follow up compared to baseline, with no significant difference between the groups.

Bleeding during menstruation decreased in both groups. The median PBAC score in the MWA group decreased from 563 to 122 and in the UAE group from 877 to 197. There was no significant difference between the groups.

**Table 1. Baseline characteristics.**

| Characteristics | MWA Median (IQR) Min-Max | UAE Median (IQR) Min-Max | p value |
|---|---|---|---|
| Age (years) | 43 (37-49) 36-51 | 46 (40-48) 36-48 | 0.816 |
| Parity | 2 (1-2) 0-4 | 2 (1-2) 1-4 | 0.468 |
| BMI | 27 (23-30) 22-32 | 26 (24-28) 23-32 | 0.728 |
| Hemoglobin | 112 (89-133) 74-136 | 128 (121-132) 104-133 | 0.07 |
| Ca-125 | 83 (50-310) 16-370 | 26 (16-40) 13-48 | **0.03** |
| AMH | 0.43 (0.24-3.6) 0.12-5.2 | 0.07 (0.06-0.4) 0.05-0.47 | 0.09 |
| Prolactin | 332 (149-818) 121-4620 | 282 (218-1071) 143-2957 | 1 |
| PBAC | 563 (295-1571) 275-2109 | 877 (420-2088) 111-2440 | 0.46 |
| SSS | 69 (60-80) 47-84 | 88 (63-100) 50-100 | 0.104 |
| QoL | 28 (12-44) 6-60 | 13 (3-24) 1-44 | 0.19 |
| Uterus size (ml) | 378 (140-529) 130-854 | 172 (149-205) 106-215 | 0.281 |

Presented as median, interquartile range and min-max values. *P*-value calculated by Mann Whitney U-test. MWA: microwave ablation; UAE: uterine artery embolization; BMI: body mass index; AMH: anti müllerian hormone; PBAC: pictorial bleeding assessment cart score; SSS: symptom severity score of the UFS-QoL questionnaire; QoL: quality of life score of the UFS-QoL questionnaire. P value < 0.05 was considered significant (bold value).

Therewere no significant differences between the groups regarding the levels of Hemoglobin, AMH, or Prolactin after treatment. Ca-125 levels decreased significantly in the UAE group and when comparing the groups in favor of UAE (p=0.035). Uterine volume (measured on MRI before and after treatment) decreased in both treatment groups but without significant differences between the groups.

Dysmenorrhea defined as NRS maximum during menstruation (on a scale from 0-10), extracted from the PBAC questionnaire, was analyzed. Both groups had a significant difference in reduction of dysmenorrhea six months after treatment. In the MWA group, the median of maximum pain reduced from NRS 6 to 1 (p=0.008) and in the UAE from 9 to 4 (p=0.02). There was no significant difference between the groups regarding treatment effect of dysmenorrhea.

**Table 2. Primary and secondary outcomes.**

| Outcome | MWA baseline | UAE baseline | MWA 6 months | UAE 6 months | MWA difference | UAE difference | p value | Median of the difference | 95% CI of the median | |
|---|---|---|---|---|---|---|---|---|---|---|
| | Median (IQR) | Median (IQR) | Median (IQR) | Median (IQR) | Median (IQR) | Median (IQR) | | | Lower | Upper |
| | Min-max | Min - max | Min - max | Min - max | Min - max | Min - max | | | | |
| | Primary outcome | | | | | | | | | |
| SSS | 69 | 88 | 44 | 47 | −30 | −9 | 1 | −22 | −31 | −9 |
| | (60–79) | (63–100) | (22–65) | (31–91) | (−37 to −21) | (−44 to −8) | | | | |
| | 47–84 | 50–100 | 19–75 | 13–91 | −53–0 | −75 to −7 | | | | |
| | Secondary outcomes | | | | | | | | | |
| HR-QoL | 27 | 13 | 79 | 67 | 56 | 50 | 0.755 | 50 | 28 | 61 |
| | (12–44) | (3–24) | (55–91) | (44–85) | (32–67) | (19–69) | | | | |
| | 6–60 | 1–44 | 34–98 | 35–88 | 15–80 | −9–85 | | | | |
| PBAC | 563 | 877 | 122 | 197 | −328 | −540 | 0.836 | −540 | −1747 | −86 |
| | (295–1571) | (420–2088) | (38–277) | (20–760) | (−1763 to −102) | (−1807–177) | | | | |
| | 275–2109 | 111–2440 | 12–414 | 6–1317 | −2028–139 | −1866–440 | | | | |
| Hb | 112 | 128 | 133 | 132 | 11 | 0 | 0.463 | 7 | 0 | 17 |
| | (89–133) | (121–132) | (99–137) | (121–136) | (5–23) | (−8–12) | | | | |
| | 74–136 | 104–133 | 95–142 | 114–140 | −3–29 | −14–17 | | | | |
| AMH | 0.4 | 0.1 | 0.6 | 0.1 | −0.2 | 0.0 | 0.165 | −0.01 | −0.24 | −0.28 |
| | (0.2 to 3.6) | (0.1 to 0.4) | (0.1 to 2.8) | (0.1 to 0.3) | (−0.9 to 0.5) | (−0.1 to 0.2) | | | | |
| | 0.1 to 5.2 | 0.1 to 0.5 | 0.1 to 6.3 | 0.1 to 0.8 | −1.0 to 1.1 | −0.2 to 0.3 | | | | |
| Ca-125 | 83 | 26 | 46 | 14 | −32 | −11 | **0.035** | −16 | −37 | −1 |
| | (50–310) | (16–40) | (23–99) | (8–37) | (−174 to −11) | (−15 to −6) | | | | |
| | 16–370 | 13–48 | 19–194 | 8–44 | −347–3 | −18 to −1 | | | | |
| Prolactin | 332 | 282 | 252 | 190 | −62 | −53 | 0.805 | −96 | −149 | 15 |
| | (149–818) | (218–1070) | (208–621) | (141–268) | (−1167–156) | (−1418 to −39) | | | | |
| | 121–4620 | 143–2957 | 193–859 | 106–296 | −4246–500 | −2689 to −37 | | | | |
| Uterus volume (mL) | 378 | 172 | 241 | 104 | 59 | 7 | 0.945 | 38 | 4 | 74 |
| | (141–53) | (149–205) | (104–378) | (96–137) | (−16–92) | (−10–70) | | | | |
| | 130–854 | 106–215 | 100–455 | 85–153 | −66–101 | −25–74 | | | | |
| Dysmenorrhea | 6 | 9 | 1 | 4 | 5 | 4 | 0.836 | 4 | 1 | 7 |
| | (3–9) | (7–10) | (0–4) | (2–9) | (0–7) | (1–6) | | | | |
| | 1–10 | 6–10 | 0–5 | 2–9 | 0–8 | 0–7 | | | | |
| Hospitalization (days) | | 0 | | 3 | | | **0.004** | 2 | 2 | 5 |
| | | (0–2) | | (2–5) | | | | | | |
| | | 0–2 | | 1–6 | | | | | | |

*(Continued)*

**Table 2.** (Continued)

| Outcome | MWA baseline | UAE baseline | MWA 6 months | UAE 6 months | MWA difference | UAE difference | p value | Median of the difference | 95% CI of the median | |
|---|---|---|---|---|---|---|---|---|---|---|
| | Median (IQR) | Median (IQR) | Median (IQR) | Median (IQR) | Median (IQR) | Median (IQR) | | | Lower | Upper |
| | Min-max | Min - max | Min - max | Min - max | Min - max | Min - max | | | | |
| Pain medication (days) | | 3 | | 10 | | | **0.029** | 5 | 2 | 14 |
| | | (0–6) | | (5–21) | | | | | | |
| | | 0–14 | | 4–42 | | | | | | |

Presented as median, interquartile range and min-max values. P-value calculated by Mann-Whitney U-test. MWA: microwave ablation; UAE: uterine artery embolization; SSS: symptom severity score of the UFS-QoL questionnaire; QoL: quality of life score of the UFS-QoL questionnaire; PBAC: pictorial bleeding assessment cart score; AMH: Anti-Müllerian hormone. P value < 0.05 was considered significant (bold values). 95% CI: 95% confidence intervals of the median of the difference.

Regarding postoperative outcomes, there was a significant difference between the groups in favor of MWA treatment: hospitalization was shorter (median 0 days compared to 3 days in the UAE group, p=0,004) and shorter time of postoperative pain medication (median 3 days compared to 10 days in the UAE group, p=0,029). Regarding exploratory outcomes, there was no significant difference between the groups in reported postoperative pain level (NRS), however a significantly quicker return to daily activities was seen in the MWA group (median 3 days for MWA and 14 days for UAE, p=0.005) (Fig 4).

There was no difference in acceptability, measured as recommendation of the treatment to a friend (score from 1-7), on day of treatment nor at follow up. All study participants could strongly recommend the treatment (score 6-7) two hours post treatment except for one participant who did not have sufficient anesthesia during UAE due to failure of epidural anesthesia (score 1). In the MWA group, one patient would not recommend the treatment to a friend (score 1-3) at follow up, compared to two patients in the UAE group. One patient in the MWA group and two in the UAE group would neither recommend nor discourage the treatment to a friend (score 4-5) at follow up. Acceptability results were analyzed by Chi 2 test due to non-numerical values, and thus not presented in Table 2.

There was one severe incident with a patient in the UAE group. Due to intense postoperative pain three days postoperatively, a computed tomography was performed, showing suspected colon diverticulum perforation. The condition was treated conservatively with antibiotics and the patient was discharged on postoperative day six. The diverticulosis was later confirmed by colonoscopy. It was considered unlikely that this condition was related to the UAE treatment, however this made evaluation of post-operative pain difficult, and she was excluded from the analyses of postoperative pain, use of analgesics and admission to hospital. One patient in the UAE group was diagnosed with and treated for cystitis postoperatively.

## Discussion

In this small, randomized pilot study of symptomatic adenomyosis patients, we had no power to identify anything but highly significant differences between groups, and the primary outcome did not differ between MWA and UAE. However, we find it of interest to report that both treatments led to symptom improvement and increased quality of life, with no report of serious adverse events related to the treatments. We are aware that multiple secondary and exploratory outcomes in a limited patient material increases the risk of false positive results.

We chose UAE as the comparative treatment to MWA as they are both low risk, uterus preserving, alternatives to more invasive surgery. We had no previous experience of these treatments for adenomyosis, and therefore chose to begin

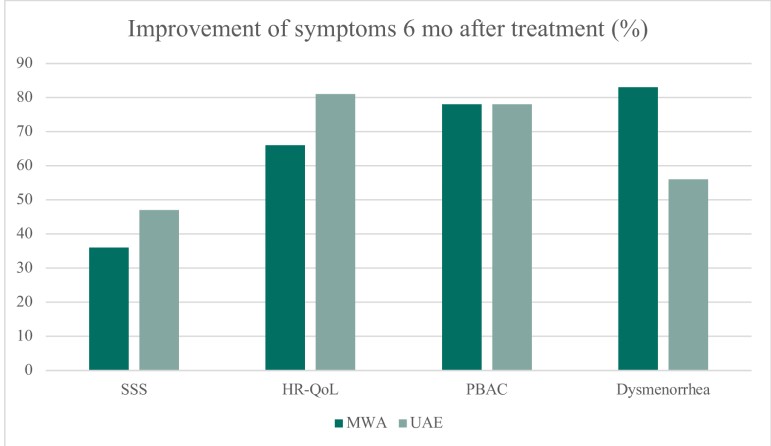

**Fig 4. Symptom alleviation in adenomyosis patients, six months after treatment with MWA or UAE (%).** Both treatments led to reduction of symptoms, but there were no significant differensces between the two groups.

evaluation of both methods in a small pilot study. At our hospital UAE was routinely performed for uterine fibroids in epidural anesthesia and MWA in general anesthesia. This could complicate immediate postoperative pain evaluation. Although patients had general anesthesia, the MWA group had shorter hospitalization, quicker return to daily activities and fewer days of postoperative pain medication. These results indicate that MWA is better tolerated postoperatively and leads to shorter absence from work compared to UAE performed in epidural anesthesia. Acceptability did not differ significantly between the groups immediately after treatment nor at follow-up. Three patients could not recommend the treatment at follow up: in the UAE group, one patient had good effect on pain but no effect on bleeding and was offered hormonal treatment, another patient (with failure of epidural anesthesia) had prolonged postoperative pain and insufficient effect on symptoms, and one patient in the MWA group had good effect on pain but continued heavy menstrual bleeding and was planned for re-ablation.

Conclusions and generalizability are severely hampered by the small sample size of this pilot trial. We chose to limit the trial population to patients without known endometriosis and/or fibroids >3cm because we wanted to assure that the treatment effect was limited to the adenomyosis condition. These strict criteria slowed inclusion of patients considerably, since adenomyosis co-exists with fibroids and/or endometriosis in 4/5 cases [3]. Inclusion mostly occurred during the Covid-19 pandemic when accessibility to health care was limited, which might have affected patient selection. In comparison to other studies, baseline symptom severity score and PBAC scores are very high in our study population with SSS of 69-88 and PBAC of 536-877 compared to other reports of preoperative SSS of 21-36 [18,31,32] and PBAC of 183 [32]. This might explain why many of our patients were satisfied with the treatments despite remaining abnormal postoperative scores.

A strength of this study is the rigorous diagnosis of adenomyosis before inclusion, with all patients being examined with both TVUS by a specialist in gynecological ultrasound as well as MRI. Patients with both focal and diffuse adenomyosis were included but we did not analyze results for these groups separately due to the small sample size. Li et al published results on MWA of 107 patients with adenomyosis in 2023 and did not find any difference of treatment effect between focal, diffuse, or mixed type at up to 12 months follow up [33].

In order to possibly assess the potential use of these treatments for women with an active desire to become pregnant, we evaluated AMH levels before and after treatment. AMH can be used as marker for functional ovarian reserve and decreases naturally with age [34]. In our study population (aged 36-51), there was no significant change in AMH in either

of the groups at follow-up, which is in accordance with previous published data [19]. This is reassuring for patients who desire future fertility. With adequate measures to avoid endometrial impairment, [35] microwave ablation could be a promising alternative for patients with a fertility wish However, the study population is too small to generalize our findings to the general populations and effects on future fertility need to be established in larger trials.

Results for this small pilot study are underpowered but indicate that both MWA and UAE can reduce adenomyosis symptoms. We find microwave ablation a feasible option with significantly shorter hospitalization and quicker return to daily activities compared to UAE, and are moving forward with a prospective interventional study on a larger, less selected patient material.

## Conclusion

Both MWA and UAE have potential as safe and acceptable treatments for adenomyosis, leading to decreased symptoms and increased quality of life, however no difference between the groups was seen regarding primary outcome. Patients in the MWA group seem to have more favorable postoperative outcome. Since this was a small pilot study; further research is needed to confirm these results and to analyze the long-term effect.

## Key message

Both MWA and UAE treatments of symptomatic adenomyosis patients resulted in symptom reduction and increased quality of life. The MWA group had more favorable postoperative recovery and no adverse events occurred.

## Supporting information

**S1 File. Consort Checklist.**
(DOC)

## Acknowledgements

The authors wish to express their gratitude to Dr. Annika Cronsioe, Dr. Charlotte Odevall and midwife Susanne Tidbeck for their engagement and contribution to this research. To statistician Fredrik Johansson with help with statistical analysis.

## Author contributions

**Conceptualization:** Klara Hasselrot.

**Data curation:** Guðný Jónsdóttir, Klara Hasselrot, Erika Lantz, Marie Beermann, Maria Paschou.

**Formal analysis:** Klara Hasselrot.

**Funding acquisition:** Guðný Jónsdóttir, Helena Kopp Kallner.

**Methodology:** Marie Beermann, Helena Kopp Kallner, Klara Hasselrot.

**Project administration:** Klara Hasselrot.

**Resources:** Helena Kopp Kallner.

**Supervision:** Helena Kopp Kallner, Klara Hasselrot.

**Validation:** Helena Kopp Kallner, Klara Hasselrot.

**Visualization:** Guðný Jónsdóttir, Klara Hasselrot.

**Writing – original draft:** Guðný Jónsdóttir, Klara Hasselrot.

**Writing – review & editing:** Guðný Jónsdóttir, Erika Lantz, Marie Beermann, Maria Paschou, Helena Kopp Kallner, Klara Hasselrot.

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
