## [Decision Letter · Decision Letter 0]

2 Jan 2026

PONE-D-25-59928Symptom improvement in adenomyosis patients after ultrasound guided microwave ablation or uterine artery embolization, a randomized controlled pilot studyPLOS One

Dear Dr. Jónsdóttir,

Thank you for submitting your manuscript to PLOS ONE. After careful consideration, we feel that it has merit but does not fully meet PLOS ONE’s publication criteria as it currently stands. Therefore, we invite you to submit a revised version of the manuscript that addresses the points raised during the review process.

We look forward to receiving your revised manuscript.

Kind regards,

Kazunori Nagasaka

Academic Editor

PLOS One

**Journal Requirements:**

1. When submitting your revision, we need you to address these additional requirements. Please ensure that your manuscript meets PLOS ONE's style requirements, including those for file naming. The PLOS ONE style templates can be found at https://journals.plos.org/plosone/s/file?id=wjVg/PLOSOne_formatting_sample_main_body.pdf and https://journals.plos.org/plosone/s/file?id=ba62/PLOSOne_formatting_sample_title_authors_affiliations.pdf 2. Thank you for stating the following financial disclosure: The study was funded by The Stockholm County Council. HKK received the funding, the sponsors were not involved in anything connected to the study.   Please state what role the funders took in the study.  If the funders had no role, please state: "The funders had no role in study design, data collection and analysis, decision to publish, or preparation of the manuscript." If this statement is not correct you must amend it as needed. Please include this amended Role of Funder statement in your cover letter; we will change the online submission form on your behalf. 3. In the online submission form, you indicated that “Data generated or analyzed during the study are available from the corresponding author on request. Data was not registered in a public database but is available at our research clinic in a local database and on paper.” All PLOS journals now require all data underlying the findings described in their manuscript to be freely available to other researchers, either a. In a public repository, b. Within the manuscript itself, or c. Uploaded as supplementary information.This policy applies to all data except where public deposition would breach compliance with the protocol approved by your research ethics board. If your data cannot be made publicly available for ethical or legal reasons (e.g., public availability would compromise patient privacy), please explain your reasons on resubmission and your exemption request will be escalated for approval. 4. Please amend either the title on the online submission form (via Edit Submission) or the title in the manuscript so that they are identical. 5. Your ethics statement should only appear in the Methods section of your manuscript. If your ethics statement is written in any section besides the Methods, please move it to the Methods section and delete it from any other section. Please ensure that your ethics statement is included in your manuscript, as the ethics statement entered into the online submission form will not be published alongside your manuscript. 6. If the reviewer comments include a recommendation to cite specific previously published works, please review and evaluate these publications to determine whether they are relevant and should be cited. There is no requirement to cite these works unless the editor has indicated otherwise.

**Additional Editor Comments:**

Dear Authors,

Thank you for submitting your manuscript. The study is well conducted and the clinical question is of interest; however, several minor statistical clarifications are still required before acceptance.

Please revise Tables 1 and 2 to report unadjusted p-values (Bonferroni correction is not necessary for this exploratory pilot study), clearly specify the statistical test used for within-group pre–post comparisons, and add summary statistics (median with IQR and range) for post–pre differences within each treatment group.

These revisions will improve transparency and interpretability of the results.

We look forward to receiving a revised version addressing these points.

Sincerely,

Kazunori Nagasaka

Plos One

Reviewers' comments:

Reviewer's Responses to Questions

**Comments to the Author**

1. Is the manuscript technically sound, and do the data support the conclusions?

Reviewer #1: Yes

Reviewer #2: Partly

2. Has the statistical analysis been performed appropriately and rigorously? 

Reviewer #1: Yes

Reviewer #2: No

3. Have the authors made all data underlying the findings in their manuscript fully available?

Reviewer #1: Yes

Reviewer #2: No

4. Is the manuscript presented in an intelligible fashion and written in standard English?

Reviewer #1: Yes

Reviewer #2: Yes

5. Review Comments to the Author

**Reviewer #1:** Thank you for the opportunity to read this manuscript.

I enjoyed reading it and particularly pleased to see focus on minimally invasive adenomyosis treatments.

Authors have addressed the earlier comments, particularly about their primary and secondary outcomes and size of the study population.

I hope this will inspire for proper RCT going forward

**Reviewer #2:**  This manuscript reports results investigating symptom improvement in adenomyosis patients after ultrasound guided microwave ablation or uterine artery embolization from a randomized controlled pilot study. I have the following comments for statistical analysis.

For Table 1, please add one column to include unadjusted p-value for each characteristic. Bonferroni correction is not needed since the tested characteristic should have been tested independently.

Page 12, lines 244 and 248, those results are pre-post comparisons within each treatment group. It is not clear what statistical test was used here. Please add appropriate description of the statistical test for pre-post comparisons in the section of Statistical analysis.

For Table 2, please report unadjusted p-value for each outcome since this study didn’t consider power estimates or effect size under given sample size. For such exploring pilot study, it is not informative with Bonferroni corrections.

In Table 2, please add two columns to report the median (IQR, and Min-max) of difference between post – pre within each treatment (MWA, or UAE) for each outcome. These data will be needed to reflect the distribution of the difference between post – pre of a studied outcome within each treatment.

6. PLOS authors have the option to publish the peer review history of their article (what does this mean? ). If published, this will include your full peer review and any attached files.

**Do you want your identity to be public for this peer review?** For information about this choice, including consent withdrawal, please see our Privacy Policy .

Reviewer #1: **Yes:** Dr Fusun Sirkeci

Reviewer #2: No

---

## [Author Response · Author response to Decision Letter 1]

5 Feb 2026

Response to Reviewers regarding PONE-D-25-59928

Dear Reviewers and Editor,

Thank you for fruitful feedback. The manuscript has now been revised according to your suggestions, and we hope you can consider it for publication.

Remark 1: The manuscript is formatted to PLOS ONE´s style requirements.

Remark 2,3 - added as suggested:

”The study was funded by The Stockholm County Council. HKK received the funding, the sponsors were not involved in anything connected to the study.”

”According to the General Data Protection Regulation (GDPR) of the European Union (EU), no data containing personal information can be shared outside the EU unless patients have consented (which the patients in the current study have not). Data can therefore not be uploaded in a repository, nor disclosed within the manuscript or uploaded as supplementary information. Data can be shared within the EU after establishment of an agreement.”

Remark 4: title on the online submission form and title in the manuscript are identical

Remark 5: ethics statement is now only apparent in the Methods section, as suggested

Remark 6: non applicable

Remark 7: non applicable

Specific remarks from reviewer#2:

Tables 1 and 2 now report unadjusted p-values and the Bonferroni correction have been deleted, as suggested by the reviewer.

The statistical test used for within-group pre–post comparisons and summary statistics (median with IQR and range) for post–pre differences within each treatment group have been clarified in the section of statistical analysis, as suggested by the reviewer:

“To compare the difference between variables that were measured before and after treatment within each group we used Paired T test and descriptive statistics.” (line 217-219)

Regarding postoperative outcomes, there was a significant difference between the groups in favor of MWA treatment: hospitalization was shorter (median 0 days compared to 3 days in the UAE group, p=0,004) and shorter time of postoperative pain medication (median 3 days compared to 10 days in the UAE group, p=0,029).

---

## [Decision Letter · Decision Letter 1]

12 Feb 2026

Symptom improvement in adenomyosis patients after ultrasound guided microwave ablation or uterine artery embolization, a randomized controlled pilot study

PONE-D-25-59928R1

Dear Dr. Jónsdóttir,

We’re pleased to inform you that your manuscript has been judged scientifically suitable for publication and will be formally accepted for publication once it meets all outstanding technical requirements.

Kind regards,

Kazunori Nagasaka

Academic Editor

PLOS One

Additional Editor Comments (optional):

Dear Dr. Jónsdóttir,

Thank you for submitting the revised version of your manuscript entitled: “Symptom improvement in adenomyosis patients after ultrasound guided microwave ablation or uterine artery embolization, a randomized controlled pilot study” (PONE-D-25-59928R1) We have now received and considered the final reviewer comments. The remaining points were minor and primarily concerned clarification of Table 2 formatting and placement of postoperative variables, which have been appropriately addressed in the revised version.

The reviewers also acknowledged that their earlier comments regarding statistical reporting, primary and secondary outcomes, and study population size were satisfactorily handled.

In particular, the reviewers appreciated the focus on minimally invasive treatment options for adenomyosis and noted that, although this is a small pilot randomized study, the work provides valuable preliminary data and may serve as an important foundation for a properly powered randomized controlled trial in the future.

After careful evaluation, I am pleased to inform you that your manuscript is accepted for publication in PLOS ONE, pending final technical checks by the editorial office.

Thank you for choosing to publish your work with PLOS ONE. We look forward to seeing your study contribute to the growing body of literature on uterus-preserving treatments for adenomyosis.

With best regards,

Kazunori Nagasaka

Academic Editor PLOS ONE

Reviewers' comments:

Reviewer's Responses to Questions

**Comments to the Author**

1. If the authors have adequately addressed your comments raised in a previous round of review and you feel that this manuscript is now acceptable for publication, you may indicate that here to bypass the “Comments to the Author” section, enter your conflict of interest statement in the “Confidential to Editor” section, and submit your "Accept" recommendation.

Reviewer #2: All comments have been addressed

2. Is the manuscript technically sound, and do the data support the conclusions?

Reviewer #2: (No Response)

3. Has the statistical analysis been performed appropriately and rigorously? 

Reviewer #2: (No Response)

4. Have the authors made all data underlying the findings in their manuscript fully available?

Reviewer #2: (No Response)

5. Is the manuscript presented in an intelligible fashion and written in standard English?

Reviewer #2: (No Response)

6. Review Comments to the Author

Reviewer #2: In Table 2, data of hospitalization and postoperative pain medication in MWA treatment should be relocated to the 4th column under MWA 6 months.

7. PLOS authors have the option to publish the peer review history of their article (what does this mean? ). If published, this will include your full peer review and any attached files.

**Do you want your identity to be public for this peer review?** For information about this choice, including consent withdrawal, please see our Privacy Policy .

Reviewer #2: No

---

## [Editor Report · Acceptance letter]

PONE-D-25-59928R1

PLOS One

Dear Dr. Jónsdóttir,

I'm pleased to inform you that your manuscript has been deemed suitable for publication in PLOS One. Congratulations! Your manuscript is now being handed over to our production team.

Kind regards,

on behalf of

Professor Kazunori Nagasaka

Academic Editor

PLOS One